# X-ray Detectors for Kaonic Atoms Research at DAΦNE

**Catalina Curceanu** [1], **Aidin Amirkhani** [2], **Ata Baniahmad** [2], **Massimiliano Bazzi** [1], **Giovanni Bellotti** [2], **Carolina Berucci** [1,†], **Damir Bosnar** [3], **Mario Bragadireanu** [4], **Michael Cargnelli** [5], **Alberto Clozza** [1], **Raffaele Del Grande** [1], **Carlo Fiorini** [2], **Francesco Ghio** [6], **Carlo Guaraldo** [1], **Mihail Iliescu** [1], **Masaiko Iwasaki** [7], **Paolo Levi Sandri** [1], **Johann Marton** [5], **Marco Miliucci** [1], **Pavel Moskal** [8], **Szymon Niedźwiecki** [8], **Shinji Okada** [7], **Dorel Pietreanu** [1,4], **Kristian Piscicchia** [1,9], **Alessandro Scordo** [1], **Hexi Shi** [1,‡], **Michal Silarski** [8], **Diana Sirghi** [1,4,*], **Florin Sirghi** [1,4], **Magdalena Skurzok** [8], **Antonio Spallone** [1], **Hideyuki Tatsuno** [10], **Oton Vazquez Doce** [1,11], **Eberhard Widmann** [5] and **Johann Zmeskal** [5]

[1]   INFN, Laboratori Nazionali di Frascati, Frascati, 00044 Roma, Italy; catalina.curceanu@lnf.infn.it (C.C.); massimiliano.bazzi@lnf.infn.it (M.B.); carolinaberucci@gmail.com (C.B.); alberto.clozza@lnf.infn.it (A.C.); raffaele.delgrande@lnf.infn.it (R.D.G.); carlo.guaraldo@lnf.infn.it (C.G.); mihai.iliescu@lnf.infn.it (M.I.); paolo.levisandri@lnf.infn.it (P.L.S.); marco.miliucci@lnf.infn.it (M.M.); dorel.pietreanu@nipne.ro (D.P.); kristian.piscicchia@gmail.com (K.P.); alessandro.scordo@lnf.infn.it (A.S.); hexishi@lnf.infn.it (H.S.); fsirghi@lnf.infn.it (F.S.); antonio.spallone@lnf.infn.it (A.S.); oton.vazquez@universe-cluster.de (O.V.D.)

[2]   Politecnico di Milano, Dipartimento di Elettronica, Informazione e Bioingegneria and INFN Sezione di Milano, 20133 Milano, Italy; aidin.amirkhani@polimi.it (A.A.); ata.baniahmad@polimi.it (A.B.); giovanni.bellotti@polimi.it (G.B.); carlo.fiorini@polimi.it (C.F.)

[3]   Department of Physics, Faculty of Science, University of Zagreb, 10000 Zagreb, Croatia; bosnar@phy.hr

[4]   Horia Hulubei National Institute of Physics and Nuclear Engineering, IFIN-HH, 077125 Magurele, Romania; mario.bragadireanu@nipne.ro

[5]   Stefan-Meyer-Institut für Subatomare Physik, Vienna 1090, Austria; michael.cargnelli@oeaw.ac.at (M.C.); johann.marton@oeaw.ac.at (J.M.); eberhard.widmann@oeaw.ac.at (E.W.); johann.zmeskal@oeaw.ac.at (J.Z.)

[6]   INFN Sez. di Roma I and Inst. Superiore di Sanita, 00161 Roma, Italy; francesco.ghio@lnf.infn.it

[7]   RIKEN, Tokyo 351-0198, Japan; masa@riken.jp (M.I.); sokada@riken.jp (S.O.)

[8]   The M. Smoluchowski Institute of Physics, Jagiellonian University, 30-348 Kraków, Poland; ufmoskal@googlemail.com (P.M.); szymonniedzwiecki@googlemail.com (S.N.); michal.silarski@uj.edu.pl (M.S.); mskurzok@gmail.com (M.S.)

[9]   Museo Storico della Fisica e Centro Studi e Ricerche "Enrico Fermi", 00184 Roma, Italy

[10]  Lund Univeristy, Faculty of Science, 22100 Lund, Sweden; hideyuki.tatsuno@gmail.com

[11]  Excellence Cluster Universe, Technische Universiät München, 85748 Garching, Germany

*   Correspondence: sirghi@lnf.infn.it

†   Current address: Physics Department, University of Rome Tor Vergata, 00133 Rome, Italy.

‡   Current address: Institute of High Energy Physics, HEPHY, Vienna 1050, Austria.

**Abstract:** This article presents the kaonic atom studies performed at the INFN National Laboratory of Frascati (Laboratori Nazionali di Frascati dell'INFN, LNF-INFN) since the opening of this field of research at the DAΦNE collider in early 2000. Significant achievements have been obtained by the DAΦNE Exotic Atom Research (DEAR) and Silicon Drift Detector for Hadronic Atom Research by Timing Applications (SIDDHARTA) experiments on kaonic hydrogen, which have required the development of novel X-ray detectors. The 2019 installation of the new SIDDHARTA-2 experiment to measure kaonic deuterium for the first time has been made possible by further technological advances in X-ray detection.

**Keywords:** kaonic atoms; strong interaction; X-ray detectors

## 1. Introduction

An exotic atom is an atomic system where an electron is replaced by a negatively charged particle, which could be a muon, a pion, a kaon, an antiproton, or a sigma hyperon, bound into an atomic orbit by its electromagnetic interaction with the nucleus.

Among exotic atoms, the hadronic ones, in which the electron is replaced by a hadron, play a unique role, since their study allows for the experimental investigation of the strong interaction described by Quantum Chromo Dynamics (QCD). The interaction is measured at threshold since the relative energy between the hadron forming the exotic atom and the nucleus is so small that it can be, for any practical purpose, neglected.

Experiments measuring kaonic atoms, in particular kaonic hydrogen, performed from the 70s through the 80s [1–3] have left the scientific community with a huge problem known as the "kaonic hydrogen puzzle": the measured strong interaction shift of the fundamental level with respect to the electromagnetic calculated value was positive, the resulting level more bound, which meant an attractive-type strong interaction between the kaon and the proton. This was in striking contradiction with the results of the analyses of low-energy scattering data, which found a repulsive-type strong interaction.

In this paper, we describe the DAΦNE Exotic Atom Research (DEAR) [4] and Silicon Drift Detector for Hadronic Atom Research by Timing Applications (SIDDHARTA) experiments [5] on kaonic hydrogen at the DAΦNE collider at the INFN National Laboratory of Frascati (Laboratori Nazionali di Frascati dell'INFN, LNF-INFN). These experiments have characterized the progress in detector development achieved in performing precision measurements of kaonic atoms.

DEAR has contributed to solve the "kaonic hydrogen puzzle", after the measurement of the KpX experiment at KEK [6], disentangling the full pattern of the *K*-series lines of kaonic hydrogen by employing charged-coupled devices (CCDs) that take advantage of their pixelized structure to obtain a powerful background reduction based on topological and statistical considerations.

SIDDHARTA used large area silicon drift detectors (SDDs) with microsecond timing capabilities. The main feature of the SDDs is the small value of the anode capacitance, enabling good resolution in energy and time. SIDDHARTA has performed the most precise measurement in the literature on kaonic hydrogen transitions.

In 2019, a new experiment, the SIDDHARTA-2 experiment, will be installed on DAΦNE to perform the first measurement of kaonic deuterium. The experimental challenge of the kaonic deuterium measurement is the yield, one order of magnitude less than kaonic hydrogen, and the even larger width. In order to satisfy the stringent requirements of the measurement, new monolithic SDD arrays have been developed with an improved technology, which increases the stability, optimizes the geometrical X-ray detection efficiency, and reduces the drift time.

Section 2 describes the kaonic atom measurements performed at DAΦNE by the DEAR experiment employing CCDs. Section 3 describes the measurements performed by the SIDDHARTA experiment employing SDDs. Section 4 looks at the future measurement of kaonic deuterium by the SIDDHARTA-2 experiment at DAΦNE. Conclusions are drawn in Section 5.

## 2. Kaonic Atom Measurements at DAΦNE Employing CCDs

### 2.1. Charge-Coupled Devices (CCDs)

Charge-coupled device (CCD) arrays are ideal detectors for a variety of X-ray imaging and spectroscopy applications, and in particular, in exotic atom research [7–9]. The CCD is essentially a silicon integrated circuit of the MOS type. The device consists of an oxide-covered silicon substrate with an array of closely spaced electrodes on top. Each electrode is equivalent to the gate of an MOS transistor. Signal information is carried in the form of electrons. The charge is localized beneath the electrodes with the highest applied potentials because the positive potential of an electrode causes the underlying silicon to be depleted to a certain depth and thus have a positive potential,

which attracts the electrons. It is therefore common to say that the electrons are being stored in a "potential well". "Charge coupling" is a technique to transfer a signal charge from under one electrode to the next (Figure 1). This is achieved by also taking the voltage of the second electrode to a high level, then reducing the voltage of the first electrode. Therefore, by sequentially pulsing the voltages on the electrodes between high and low levels, charges can be made to pass down an array of many electrodes with hardly any loss and very little noise.

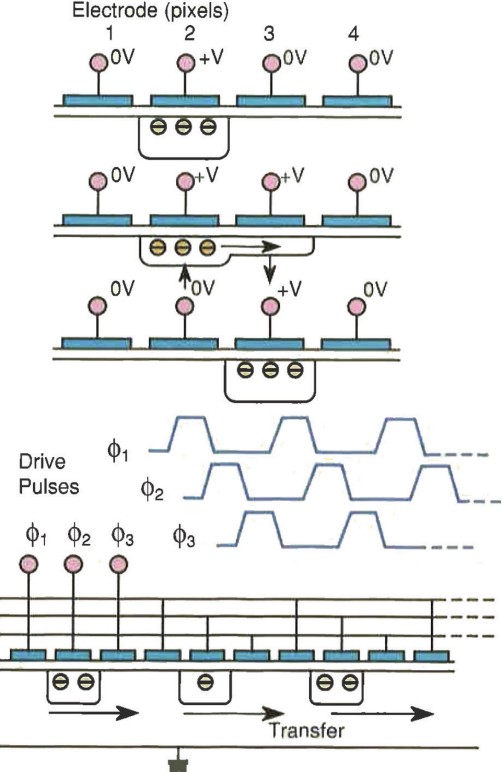

**Figure 1.** Charge signal transfer from one pixel to the next. One pixel contains three electrodes. The charges are in electrode No. 2, which has a voltage +V of 10 V. The voltage of electrode No. 3 is set to the same level as that of No. 2. Simultaneously, the voltage of No. 2 is reduced and the charges move to No. 3. A sequential pulsing of electrodes between two levels therefore allows a charge signal transfer over many electrodes (pixels).

CCDs are operated in vacuum and cooled down to approximately 160 K in order to limit dark current and therefore allow for up to several hours of exposure time. They operate in a similar way to conventional silicon solid state detectors in that the incoming X-rays, following absorption by photoelectric effect, are converted to electron–hole pairs where each pair requires 3.68 eV for its creation. In contrast to the visible photon case, the number of electrons created depends on the X-ray energy, and a good energy resolution can therefore be achieved.

The energy resolution of a CCD is given by:

$$\Delta E \ FWHM(eV) = 2.355 \times 3.68(N^2 + \frac{FE}{3.68})^{1/2}, \tag{1}$$

where $N$ is the r.m.s. transfer and readout noise of the CCD, $F$ is the Fano factor, and $E$ is the X-ray energy. From the formula above, the best possible energy resolution with Si CCDs can be estimated by considering $N^2$ very small. The result is 70 eV FWHM at 2 keV and 140 eV FWHM at 8 keV. The characteristic parameters of CDDs are reported in Table 1.

**Table 1.** Comparison of X-ray detectors for kaonic atom research.

| Detector | Si(Li) | CCD | SDD-JFET | SDD-CUBE |
|---|---|---|---|---|
| Effective area (mm$^2$) | 200 | 724 | $3 \times 100$ | $8 \times 64$ |
| Thickness (mm) | 5 | 0.03 | 0.45 | 0.45 |
| Energy resolution, FWHM, (eV) at 6 keV | 410 | 150 | 160 | 140 |
| Drift time (ns) | 290 | - | 800 | 300 |
| Experiment | KpX | DEAR | SIDDHARTA | SIDDHARTA-2 |
| Reference | [6] | [4] | [5] | [10] |

The identification of X-ray events and the determination of their energies is achieved by taking advantage of the pixel structure, which allows the application of a selection based on topological and statistical criteria [11]. This powerful background rejection tool is based on the fact that X-rays in the 1–10 keV energy range interact mainly via photoelectric effect and have a high probability of depositing all their energy in a single, or at most two, adjacent pixels, whereas the energy deposited from background particles (charged particles, gammas, neutrons) is distributed over several pixels (therefore called "cluster events"), which can be rejected (see Figure 2 (left)). A selected "single-pixel" (see Figure 2 (right)), a pixel with a charge content above a selected noise threshold, that is surrounded by eight neighbor pixels having a charge content below that threshold is considered to be an X-ray hit.

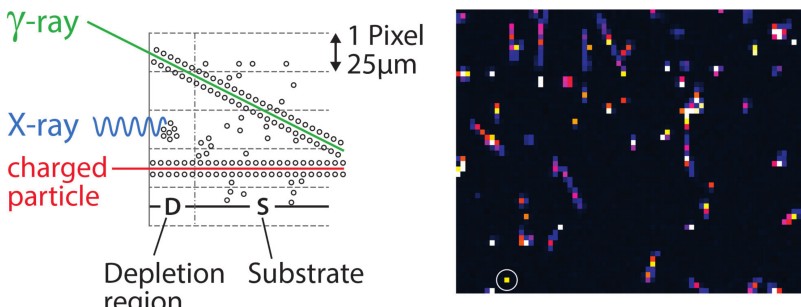

**Figure 2.** (**left**) Particle interactions and charge collection in a CCD detector; (**right**) example of an X-ray signal (inside the circle) in a CCD picture exposed during a data taking run.

### 2.2. The DEAR Kaonic Hydrogen Measurement at DAΦNE

After the KEK result, the primary goal of the DEAR experiment at the LNF-INFN $e^+e^-$ DAΦNE collider was a precise determination of shift and broadening, due to strong interaction, of the fundamental level of kaonic hydrogen and a complete identification of the pattern of lines of the *K*-series transitions. The DEAR experiment took advantage of the clean (no contaminating particles in the beam), low-momentum (127 MeV/c), nearly monoenergetic ($\Delta p/p = 0.1\%$) beam of kaons from the decay of $\phi$-mesons produced by $e^+e^-$ collisions in the DAΦNE collider.

A shaped degrader of Kapton foils, from 150 μm up to 1200 μm thickness was used to put kaons at rest in the hydrogen atoms. The stopping efficiency was about 1%, with an intrinsic efficiency of almost 100%. At KEK, the 600 MeV kaon beam was produced by the 16 GeV proton beam of the KEK Proton Synchrotron on a thick target and then brought to rest in the hydrogen target using graphite degrader a few tens of cm thick. Due to the production mechanism, kaons were accompanied by pions in a ratio K/$\pi$ equal to 1/90. The stopping efficiency was 0.06%, with an intrinsic efficiency of about 2% due to the broad energy distribution.

The DEAR setup consisted of three components: a kaon detector, a cryogenic target system, and an X-ray detection system. Figure 3 shows a schematic view of the setup. The whole setup was installed in one of the two interaction regions of DAΦNE.

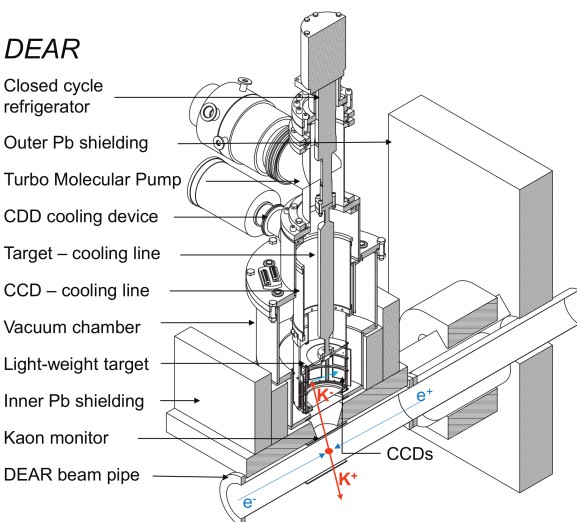

**Figure 3.** Schematic view of the DEAR experimental setup; only the right outer lead wall shielding is shown.

For X-ray detection, Marconi Applied Technologies CCD55-30 chips were selected. Each CCD55-30 chip has $1152 \times 1242$ pixels of $22.5 \times 22.5$ $\mu m^2$, resulting in a total effective area of $7.24$ $cm^2$ per chip. The depletion depth is about 30 $\mu m$. The study of transport and charge integration procedures has shown that the best results in terms of resolution and linearity can be obtained with a readout time of about 90 s. The evaluation of the occupancy effect indicates that a total exposure (readout plus static) of 120 s does not significantly reduce efficiency. Since the amount of data to be collected for each readout was relatively high, a period of 2 min was chosen. During the readout, the CCDs are exposed and since no imaging was necessary, the whole acquisition could be done in continuous readout (no static exposure). The target cell was surrounded by 16 CCDs covering a total area of 116 $cm^2$ and facing the cryogenic target cell. The CCD front-end electronics and controls and the data acquisition system were specially made for this experiment.

The number of hit pixels in a cluster categorizes the event type. In the DEAR analysis [11], events having 1 or 2 hit pixels were selected as X-ray events to increase both X-ray detection efficiency and the signal-to-noise ratio. The typical fraction of hit pixels per frame was about 3–5% so as to have an efficiency of hit recognition of about 98–99%. The X-ray detection efficiency as a function of energy and the X-ray event loss due to pile-up effect were calculated by means of Monte Carlo simulations and laboratory tests. The effect of applying charge-transfer efficiency corrections was an improvement in the resolution from 214 eV (FWHM) to 176 eV at the $K_\alpha$ line of Cu (8040 eV).

An energy calibration procedure based on fluorescence lines from setup materials excitation and the Ti and Zr foils was applied for each detector. Data from all individual detectors were then added. The overall resolution of the sum of detectors was determined: the values range from 130 eV (FWHM) for Ca $K_\alpha$ (3.6 keV) to 280 eV for Zr $K_\alpha$ (15.7 keV). The energy spectra consist of a continuous background component, fluorescence lines from setup materials, and kaonic hydrogen lines.

A measurement with non-colliding beams, i.e., $e^+e^-$ beams separated in the interaction region, was performed. These data represented the so-called "no collisions background".

Two independent analyses were performed. The two analysis methods differ essentially in the background spectrum used. Analysis I used the bulk of no collisions data as the background spectrum. Analysis II used as the background spectrum the sum of kaonic nitrogen data [12], taken initially in order to optimize the kaon stopping distribution and to characterize the machine background, and a subset (low CCDs occupancy) of no collisions data. The two analyses gave consistent results. Figure 4 shows the kaonic hydrogen X-ray spectra for both analyses after continuous and structured background subtraction.

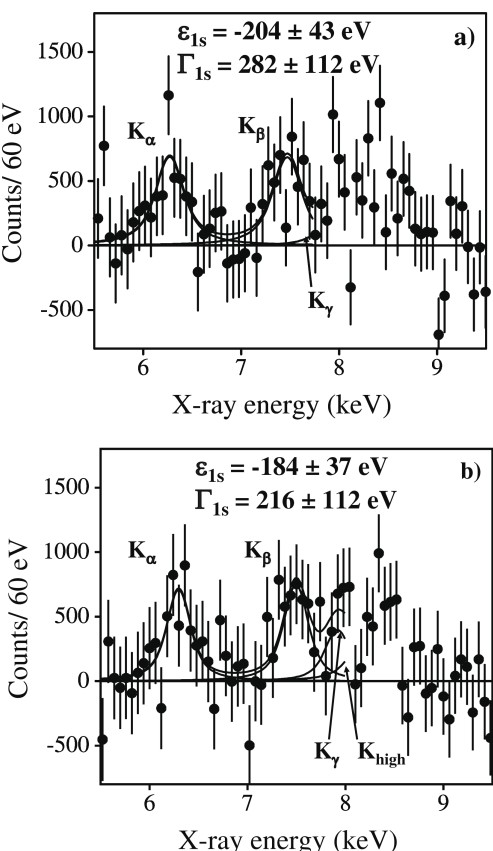

**Figure 4.** The DEAR kaonic hydrogen X-ray spectrum after continuous and structured background subtraction: (**a**) results of analysis I; (**b**) results of analysis II. The fitting curves of the various kaonic hydrogen lines are shown [4].

The resulting weighted average of the ground state shift $\varepsilon_{1s}$ was

$$\varepsilon_{1s} = -193 \pm 37(stat) \pm 6(syst) \ eV. \tag{2}$$

The weighted $1s$ ground state width $\Gamma_{1s}$ was

$$\Gamma_{1s} = 249 \pm 111(stat) \pm 30(syst) \ eV. \tag{3}$$

The DEAR results were consistent with the KEK measurement [6] to within $1\sigma$ of their respective errors. The repulsive-type character of the $K^-p$ strong interaction was confirmed.

The uncertainty of the DEAR results was about twice smaller than that of the KpX values. DEAR observed the full pattern of kaonic hydrogen $K$-lines, clearly identifying the $K_\alpha$, $K_\beta$, and $K_\gamma$ lines. The statistical significance of the summed intensities of the $K$-lines was 6.2 $\sigma$.

## 3. Kaonic Atom Measurements at DAΦNE Employing SDDs

### 3.1. Silicon Drift Detectors (SDDs)

The silicon drift detector, in its basic form proposed by Gatti and Rehak [13–15] in 1983, is a fully depleted detector in which an electric field parallel to the surface, created by properly biased contiguous field strips, drives signal charges towards a collecting anode (see Figure 5). The unique feature of this detector is the extremely low anode capacitance, which is moreover independent of the detector area. To take full advantage of the low output capacitance, the front-end n-channel JFET is integrated on the detector chip close to the $n^+$ implanted anode (Figure 5). They are located

on the upper side of the device in the center of the $p^+$ field rings. Thus, stray capacitances of the various connections are minimized and a correct matching between detector and front-end electronics capacitance can be achieved.

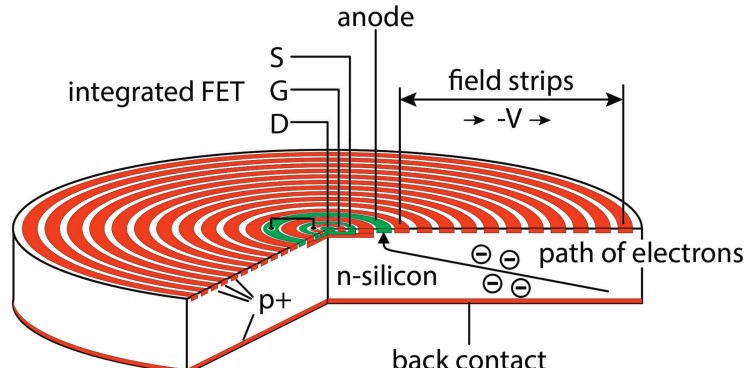

**Figure 5.** Cross section of a cylindrical silicon drift detector with integrated n-channel JFET. The gate of the transistor is connected to the collecting anode. The radiation entrance window for the ionizing radiation is the non-structured backside of the device.

The n-type device substrate is fully depleted by applying a negative voltage (with respect to the collecting anode) to the $p^+$ back contact and the $p^+$ field strips on the opposite side (see Figure 5). On this side, the negative bias of the $p^+$ rings progressively increases from the ring next to the anode to the farthest, outermost one. The maximum negative voltage of the outermost ring is about two times the voltage of the back contact. The minimum potential energy for electrons falls diagonally from the backside edge of the device to the readout electrode in the center of the upper side. Each electron generated inside the depleted detector volume by the absorption of ionizing radiation will therefore drift to the $n^+$ readout node. The generated holes are collected by the reverse-biased $p^+$ implanted regions.

As the device is fully depleted, the total thickness of 450 μm is sensitive to the absorption of ionizing radiation. For X-rays, this allows for more than 90% detection efficiency at 10 keV and more than 50% at 15 keV.

### 3.2. The SIDDHARTA Kaonic Hydrogen Measurement at DAΦNE

The SIDDHARTA experiment on DAΦNE at LNF-INFN [5] aimed to determine the kaonic hydrogen 1*s* shift and width with a higher precision than in DEAR [4], using large area SDDs.

Figure 6 shows a schematic view of the SIDDHARTA setup, which consisted of three main components: the kaon detector, X-ray detection system, and a cryogenic target system.

The SDDs in the SIDDHARTA experiment were developed within a European research project devoted to this experiment. Each of the 144 SDDs used in the apparatus had an area of 1 cm$^2$ and a thickness of 450 μm. Three cells were packed monolithically in one unit, as shown in Figure 7. The SDDs, operated at a temperature of ∼170 K, had an energy resolution of 183 eV (FWHM) at 8 keV and a timing resolution below 1 μs, in contrast to the CCD detectors used in DEAR which had no timing capability. The characteristic parameters of the SDDs used by SIDDHARTA are reported in Table 1.

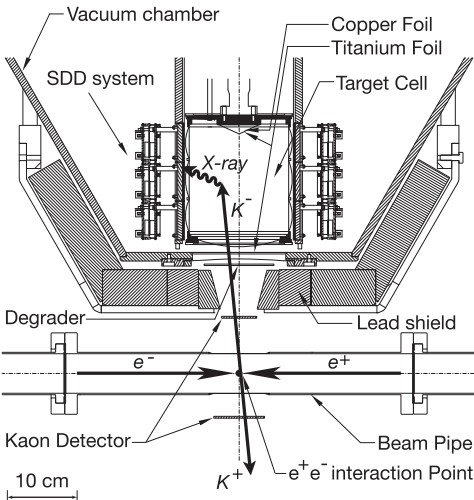

**Figure 6.** Schematic view of the SIDDHARTA setup [16].

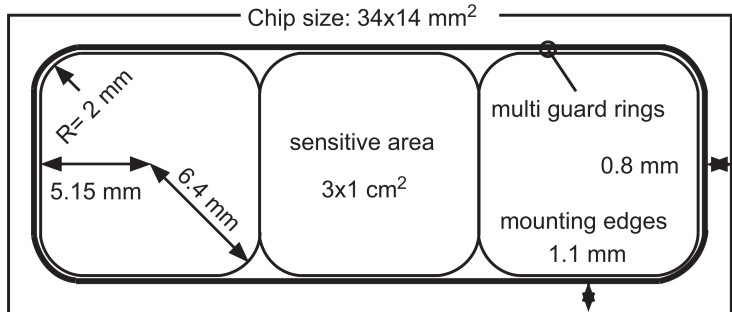

**Figure 7.** Schematic image of the SIDDHARTA SDDs. Each cell has an active area of 1 cm². Three cells are packed monolithically in one unit.

A trigger condition was used which took advantage of the characteristics of the back-to-back correlated charged kaon was production at DAΦNE, The time resolution of the SDDs allowed for the detection of the kaonic X-rays in coincidence with the back-to-back correlated $K^+ K^-$ pairs. There are two background sources in DAΦNE: backgrounds synchronous and asynchronous to the $K^+ K^-$ production. The main source is an asynchronous background and is due to electromagnetic showers originating from $e^+ e^-$ losses by the Touschek effect [17] and the interaction of the beams with the residual gas. The synchronous background, originating from particles produced by $\varphi$ decay and secondary particles produced by kaon reactions as well as the decay particles of kaons, is small. Therefore, events related to charged-kaon production are selected only by demanding a triple coincidence of K$^+$, K$^-$ and X-ray signal, so that the asynchronous background is rejected.

Figure 8 shows the time difference between the coincidence signals of the kaon monitor and the SDD events. The peak region contains the kaon-induced signals (kaonic atom X-rays) and background (gamma-rays and charged particles from the $K^-$ interactions and $K^+$ decays). The tail of the distribution indicates the charge drift time in the SDDs. The time window indicated by the thick arrows was selected as synchronous events with charged kaons. The width of the timing window, from 2.4 μs to 4.6 μs, was adjusted to maximize the signal-to-background ratio and the statistical precision of the determined kaonic atom X-ray energy.

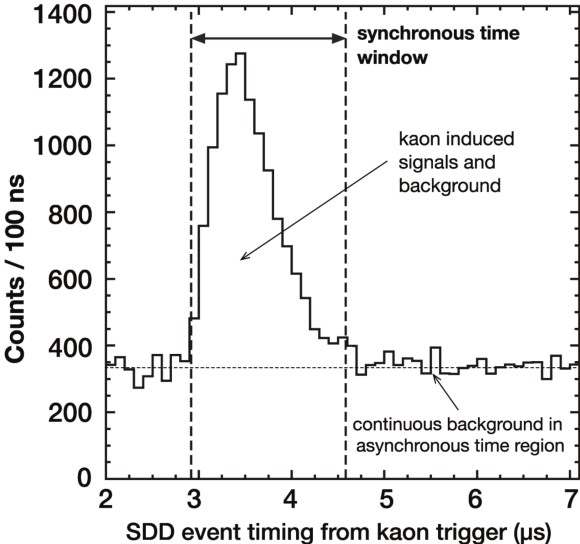

**Figure 8.** Timing spectrum of SDDs. The spectrum refers to a dataset with a target filled with helium-3. The peak corresponds to the time difference between the coincidence of the two scintillators of the kaon monitor and the X-ray events in the SDDs (triple coincidence). The peak region contains the kaon-induced signals and background. The time window indicated by arrows was selected to identify synchronous events with charged kaons. The continuous asynchronous background was reduced by this timing cut. From [16].

Using the coincidence between $K^+K^-$ pairs and X-rays measured by SDDs, the main source of asynchronous background was drastically reduced, eventually resulting in an improvement of the signal-to-background ratio by more than a factor 10 with respect to the corresponding DEAR ratio of about 1/100.

The data acquisition system was built on a PCI bus base. The differential output signal from the readout chips was read out by ADC modules. Chip control, management of the memory and timing information, and the event construction were processed by FPGA modules. Energy data for all of the X-ray signals detected by the SDDs were recorded. In addition, a time difference between a trigger signal (generated by the coincidence signals in the kaon detector) and an X-ray signal in the SDDs was recorded using a clock with a frequency of 120 MHz whenever the coincidence signals occurred within a time window of 6 μs. This timing difference information is included inside the SDD data, which are used for the selection of kaon-timing events.

In the beginning of the SIDDHARTA runs, stability checks of SDD performance were examined using the kaonic helium X-ray lines by installing a thin Ti foil and a $^{55}$Fe source inside the setup [18]. In Figure 9a, the peak position of the Mn $K_\alpha$ line (5.9 keV) as a function of time (about two weeks) is plotted, where the origin of the vertical axis is taken as an average of the Mn $K_\alpha$ peak positions in the whole dataset. A stability within ±2–3 eV was measured. This small instability was corrected to a fluctuation of ±0.5 eV in the data analysis, as indicated by "with correction" in the figure. In Figure 9b, the Mn $K_\alpha$ peak position against hit rates of the SDDs is plotted, where the origin of the vertical axis is taken as an average of the peak positions, and the horizontal axis is given by an arbitrary unit. The peak shift caused by hit rate dependency was found to be about ±2 eV, but this rate dependency was corrected from the relation between the rate and peak shift. With this correction, the rate dependency was corrected to be within ±0.5 eV. This stability is enough to determine the X-ray energy of the kaonic atom X-rays within the goal of the measurements.

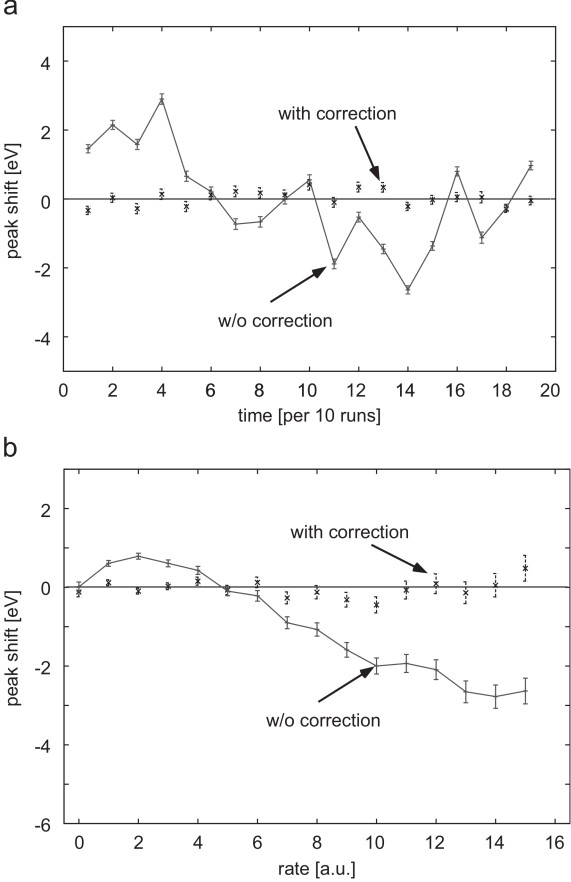

**Figure 9.** The X-ray peak shifts of the Mn K $_\alpha$ line (5.9 keV) as a function of time (**a**) and rate (**b**). The origin of the horizontal axis in the figures is the average of the data. With correction of the time dependency and rate dependency, a stability of $\pm 0.5$ eV was found [18].

In order to sum up the individual SDDs, the energy calibration of each single SDD was performed by periodic measurements of fluorescence X-ray lines from titanium and copper foils excited by an X-ray tube, with the $e^+e^-$ beams in kaon production mode. A remote-controlled system moved the kaon detector out and the X-ray tube in once every 4 h for these calibration measurements. The refined in-situ calibration in gain (energy) and resolution (response shape) of the summed spectrum of all SDDs was obtained using titanium, copper, and gold fluorescence lines excited by the uncorrelated background without trigger and also using the kaonic carbon lines from wall stops in the triggered mode.

The use of the kaonic deuterium spectrum turned out to be essential to quantify the background lines originating from kaons captured in elements such as carbon, nitrogen, and oxygen contained in the setup materials, the deuterium data having no peak structures of $K^-$d X-rays due to their low yields and broad natural widths.

A global simultaneous fit of the hydrogen and deuterium spectra was performed. Figure 10a shows the residuals of the measured kaonic hydrogen X-ray spectrum after subtraction of the fitted background. *K*-series X-rays of kaonic hydrogen were clearly observed, while those for kaonic deuterium were not visible [5]. Figure 10b,c shows the fit result with the fluorescence lines from the setup materials and a continuous background. The vertical dot-dashed line in Figure 10 indicates the X-ray energy of kaonic hydrogen $K_\alpha$ calculated using the electromagnetic interaction only. When comparing the measured kaonic hydrogen $K_\alpha$ peak with the electromagnetic value, a repulsive-type shift (negative $\varepsilon_{1s}$) of the 1$s$ energy level resulted.

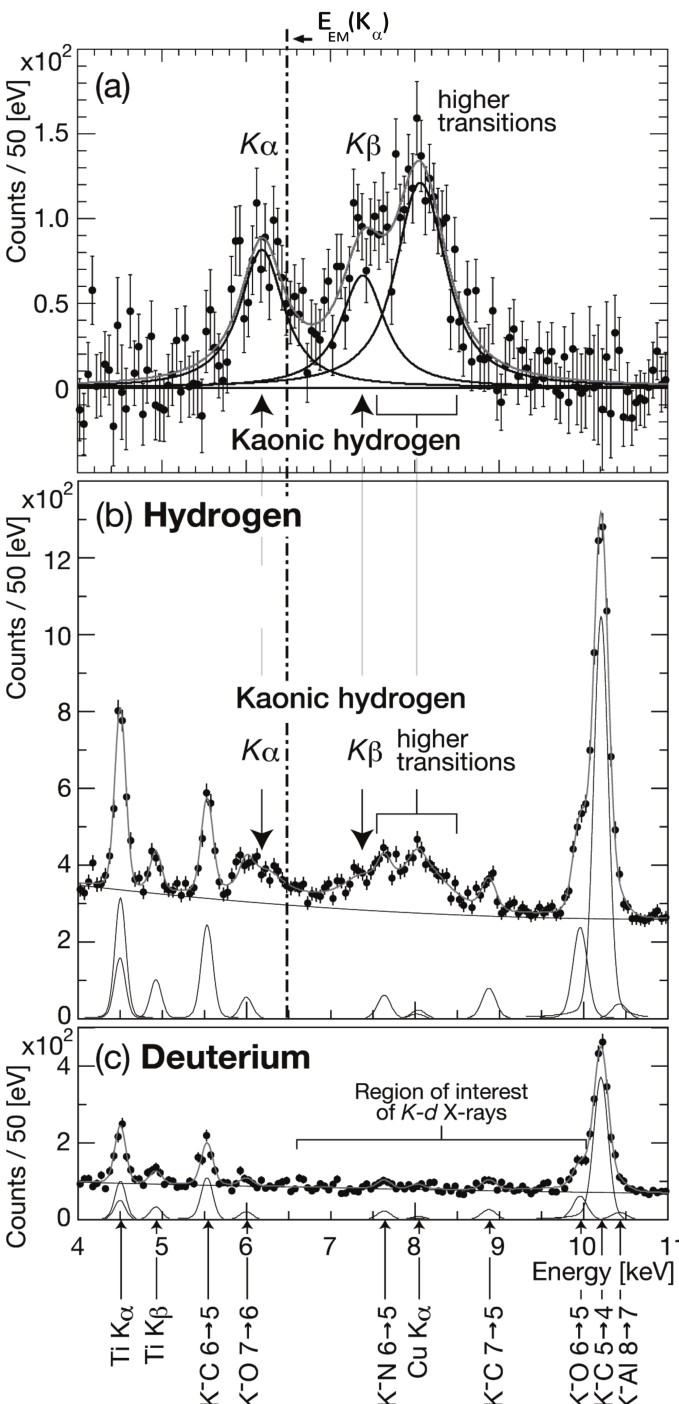

**Figure 10.** The global simultaneous fit of the X-ray energy spectra of hydrogen and deuterium data. (**a**) Residuals of the measured kaonic hydrogen X-ray spectrum after subtraction of the fitted background, clearly displaying the kaonic hydrogen *K*-series transitions. The fit components of the $K^-p$ transitions are also shown, where the sum of the functions is drawn for the higher transitions (greater than $K_\beta$). (**b,c**) Measured energy spectra with fit lines. Fit components of the background X-ray lines and a continuous background are also shown. The dot-dashed vertical line indicates the e.m. value of the kaonic hydrogen $K_\alpha$ energy [5].

The 1$s$-level shift $\varepsilon_{1s}$ and width $\Gamma_{1s}$ of kaonic hydrogen were determined to be

$$\varepsilon_{1s} = -283 \pm 36(stat) \pm 6(syst) \; eV, \tag{4}$$



$$\Gamma_{1s} = 541 \pm 89(stat) \pm 22(syst) \, eV. \tag{5}$$

This is the most precise measurement of X-rays from the kaonic hydrogen atom performed so far.

## 4. Future Measurements on DAΦNE: Kaonic Deuterium

The kaonic deuterium X-ray measurement represents the most important experimental information missing in the low-energy antikaon–nucleus interactions field.

The experimental challenge of the kaonic deuterium measurement is the very small kaonic deuterium X-ray yield of one order of magnitude less than for hydrogen and the even larger width. There are two conditions which have to be fulfilled for the kaonic deuterium measurement at DAΦNE:

- A large area X-ray detector with good energy and timing resolution and stable working conditions. To meet the stringent requirements, new monolithic SDDs arrays have been developed. A difference with respect to the previously used SDDs is the change in the pre-amplifier system from a JFET structure on an SDD chip to a CMOS integrated charge-sensing amplifier (CUBE) [19]. For each SDD cell, this CUBE amplifier is placed on the ceramic carrier as close as possible to the anode of the SDD. The anode is electrically connected to the CUBE with a bonding wire. This makes the SDDs' performance almost independent of the applied bias voltages and increases their stability, even when exposed to high charged particle rates. A better drift time of 300 ns can be achieved with the newly developed SDDs compared to the previous ones ($\sim$800 ns) by changing the active cell area from 100 mm$^2$ to 64 mm$^2$ and by further cooling to 100 K. A new readout ASIC, named SFERA, has been developed to read out the SDDs of the SIDDHARTA-2 experiment [20]. The characteristic parameters of the SDDs used for SIDDHARTA-2 are reported in Table 1.
- Dedicated veto systems, to improve the signal-to-background ratio by at least one order of magnitude as compared to the kaonic hydrogen measurement performed by SIDDHARTA. Two special veto systems are foreseen for SIDDHARTA-2, consisting of an outer barrel of scintillator counters read by photomultipliers (PMs) and called Veto-1, and an inner ring of plastic scintillation tiles (SciTiles) read by silicon photomultipliers (SiPMs) placed as close as possible behind the SDDs for charged particle tracking, called Veto-2.

## 5. Conclusions

The experimental challenge in measuring kaonic atoms consists in the need to extract a weak signal under the high background conditions of the accelerators delivering kaon beams. This has required a continuous advance in X-ray detection that characterizes the era of precision measurements.

The first kaonic hydrogen X-ray measurement, which started the modern era of kaonic atoms research, made use of Si(Li) detectors in the KpX experiment at KEK (Japan). Charge-coupled devices (CCDs) were successfully used as X-ray detectors for the DEAR experiment at LNF-INFN. Finally, silicon drift detectors (SDDs) were developed for the SIDDHARTA program at DAΦNE. R&D work on SDDs continued, leading to an optimized detector for the future kaonic deuterium program at LNF-INFN. A comparison of the main characteristics of these detectors is given in Table 1.

**Author Contributions:** Conceptualization, C.C., C.G., J.M., M.I. (Mihail Iliescu), M.I. (Masaiko Iwasaki); software, C.B., M.B., M.C., R.D.G., M.I., M.M., S.O., D.P., K.P., H.S., D.S. and H.T.; methodology, A.A., A.B., M.B. (Massimiliano Bazzi), G.B., M.B. (Mario Bragadireanu), A.C., D.B., C.F., F.G., M.I., M.M., P.M., S.N., S.O., D.P., A.S., F.S., M.S. (Mical Silarski), M.S. (Magdalena Skurzok), H.T., O.V.D., E.W. and J.Z.; writing—original draft preparation, C.C. and C.G.; writing—review and editing, C.C. and D.S.

**Funding:** This research was funded by the Austrian Science Fund (FWF), P24756-N20; the Austrian Federal Ministry of Science and Research (BMBWK), 650962/0001 VI/2/2009; the Grant­in-Aid for Specially Promoted Research (20002003), MEXT, Japan; the Croatian Science Foundation under project IP-2018-01-8570; the Minstero degli Affari Esteri e della Cooperazione Internazionale, Direzione Generale per la Promozione del Sistema Paese (MAECI), Strange Matter Project; The Polish National Science Center, through grant No. UMO-2016/21/D/ST2/01155; and the Ministry of Science and Higher Education of Poland, grant no 7150/E-338/M/2018.

**Acknowledgments:** We thank C. Capoccia and G. Corradi from LNF-INFN, and H. Schneider, L. Stohwasser, and D. Pristauz-Telsnigg from the Stefan Meyer Institute for their fundamental contribution in designing and building the SIDDHARTA setup. We also thank the DAΦNE staff for the excellent working conditions and constant support.

**Conflicts of Interest:** The authors declare no conflict of interest.

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
