# Peer review of "X-ray Detectors for Kaonic Atoms Research at DAΦNE"

_condensedmatter, doi:10.3390/condmat4020042_

Reviewer 1 Report

The key-word for accepting this manuscript is a REVIEW article. To the best of my knowledge all the results in this article have been presented over more than a decade in  specialized and detailed reports, as seen from the list of references. However, as a special issue of 'Condensed Matter' it is most

likely intended for a readership that is not usually exposed to the topic of exotic atoms. With that in mind I suggest the following:

Section 3.1: Add a brief description of the technological improvements in the KEK experiment, compared to the three 'historical' experiments.

Section 3.2: Explain in a few additional words why DAPHNE is unique in its ability to provide low energy K^+ and K^- beams in coincidence.

That will explain why much detector development work has been done around these experiments.

Section 3.3: add a reference to the Touschek effect (line 180).

Author Response

Section 3.1

This Section has been suppressed since it refers to a physics case outside of the scope of a review dedicated to detectors developments.

Section 3.2:  

A paragraph has been added showing the uniqueness of DAFNE.

Section 3.3:

A reference for the Touschek effect has been added.

Reviewer 2 Report

The paper "X-ray detectors for kaonic atoms research" gives a nice overview of the results on kaonic atoms, explaining well the current status and outlook for the future. However, from the title, one would expect a paper strongly dedicated to detectors, which is in reality not the case. Detectors are described only generally for a total length of 25% of the paper. The rest of the paper gives a review of the current kaonic atom research status. Only results already published are presented (well referenced). A deeper detector description would not only link the paper to the topic of the special issue,  but also give some uniqueness to this paper, that otherwise lacks.

For example, one could add details about the calibration of the detectors, quantify rate expected and comment on the detector performance.    

Author Response

We have performed a deeper detector description in order to have a paper more specifically dedicated to detectors. We have given technical details on detector performances. At the same time we have suppressed paragraphs referring to physics results outside the scope of a review focussed on detector developments.

Rou

nd  2

Reviewer 1 Report

This version is better balanced than the previous one.

Publication is recommended.

Author Response

Thank you very much for your support!

Reviewer 2 Report

This paper presents the  status of Xray detectors for kaonic atoms searches. This second revision appears better structured as now a larger portion of the text is actually dedicated to detectors.

There are a few minor fixes required:
- Figure 1: The figure appears too large compared to the other figures in the paper. Resize it a bit.
- Equation 1 p.4 : change 2.35 -> 2.355 as at line 69 you quote 2.355
- l.69. The conversion energy for e/hole pair has already been introduced at l.64. Remove redundancy
-l.92: ”a few tens cm thick” sounds strange, possibly “a few  tens of cm thick”
-l. 100: add a square to the dimension: 22.5 x 22.5 um^2
-l. 100 missing space: 7.24 cm^2 (space) per chip
-l.155 there-fore -> therefore
-l. 164: SDDs defined already at l.32 and l.138
-l. 181: insert space: small.(space)Therefore
-l. 203: insert space: setup.(space)In
-l.204: there is a single Mn K_alpha position traced over time: “the peak position of the Mn…..is plotted”. The same for the following lines.
l.206 and following: it is not clear how you correct the instability
l. 216: insert a space “mode.(space)A”
l.243 : SDD already defined  several times

Author Response

Fig. 1 has been revized.

All corrections suggested have been made.

Line 206: "it is not clear how you correct the stability".
The reference where the analysis is described was given.